# Antioxidant Properties and Proximate Composition of Different Tissues of European Beaver

**DOI:** 10.3390/molecules27248973

**Published:** 2022-12-16

**Authors:** Mariusz Florek, Piotr Domaradzki, Piotr Skałecki, Małgorzata Stryjecka, Katarzyna Tajchman, Agnieszka Kaliniak-Dziura, Anna Teter, Monika Kędzierska-Matysek

**Affiliations:** 1Department of Quality Assessment and Processing of Animal Products, University of Life Sciences in Lublin, Akademicka 13, 20-950 Lublin, Poland; 2The Institute of Human Nutrition Science and Agricultural, University College of Applied Sciences in Chełm, Pocztowa 54, 22-100 Chełm, Poland; 3Department of Animal Ethology and Wildlife Management, University of Life Sciences in Lublin, Akademicka 13, 20-950 Lublin, Poland

**Keywords:** *Castor fiber*, muscle tissue, adipose tissue, phenolics, antioxidant capacity, retinol, tocopherol

## Abstract

The chemical composition, content of cholesterol, retinol and α-tocopherol, and the total antioxidant capacity of different tissues from wild beavers were investigated. The total phenolic contents and free radical scavenging activity (DPPH and ABTS assays) were analysed spectrophotometrically, and fat-soluble vitamins were quantified using high-performance liquid chromatography. The type of tissue (skeletal muscle from loin and hind leg vs. adipose tissue from subcutaneous fat and tail) significantly affected content of all chemical components. The concentration of cholesterol was not related to total fat content. The retinol and α-tocopherol contents (µg/100 g) were significantly higher in the tail (13.0 and 391.2) and subcutaneous fat (12.2 and 371.3) compared to skeletal muscles (as an average 9.1 and 361.4). Among all tissues the tail showed significantly the highest values of DPPH (3.07 mM TE/100 g), ABTS (3.33 mM TE/100 g), and total phenolics (TPC, 543.7 mg GAE/100 g). The concentration of retinol was positively correlated with α-tocopherol (0.748, *p* < 0.001), and both vitamins were very strongly correlated with DPPH (0.858 and 0.886, *p* < 0.001), ABTS (0.894 and 0.851, *p* < 0.001), and TPC (0.666 and 0.913, *p* < 0.001). The principal component analysis proved that moisture, ash, and protein contents were representative for skeletal muscles, whereas, retinol, α-tocopherol, ABTS and DPPH accurately described the antioxidant capacity of tissue from the tail.

## 1. Introduction

Currently, meat from hunted game is recognised as a food of very high nutritional value [1,2] with potentially high bioactivities [3] that can successfully supplement a diet [4] or even be a primary source of animal protein in human food [5]. Game animals live in the wild and feed exclusively on natural food, or are fed by humans in times of nourishment insufficiency (e.g., in winter). Therefore, game meat does not contain hormonal substances, antibiotic residues, additives, or other compounds that may be reasonably expected in the meat of domestic animals for slaughter.

One of the main chemical reactions decreasing the quality, shelf life, and acceptability of meat and meat products is oxidation of lipids [6]. Plants and plant materials contain a high proportion of natural bioactive substances acting as antioxidants [7]. Therefore, meat from animals whose diet is based on natural plants shows a long shelf live (oxidative stability) [8] owing to the presence in animal tissues of a complex of endogenous antioxidant compounds of a hydro- and lipophilic nature [9], which counteract oxidative stress [10]. However, the total oxidative stability of muscle tissue (as a complex matrix) depends on the amount of and mutual relations between prooxidants and antioxidants [11]. The mechanism of lipid oxidation can occur via non-enzymatic (auto- and photo-oxidation) or enzymatic (involving lipoxygenase) pathways [12]. In the case of muscle tissue, auto-oxidation (peroxidation) is the most important process of lipid oxidation [13].

The content of antioxidant compounds and oxidative stability were investigated in meat from domestic [14,15,16] and wild [17,18] animals. The nonenzymatic antioxidant compounds include primarily vitamin E homologues (tocochromanols: tocopherols and tocotrienols) [19], vitamin C (L–ascorbic acid), carotenoids (carotenes and xanthophylls), and phenolic and polyphenolic compounds [7]. The enzymatic antioxidant system mostly included superoxide dismutase, catalase, and glutathione peroxidase [20]. Together they can act effectively on many levels as primary and secondary antioxidants (as scavengers, quenchers, inactivators, chelators, inhibitors, and reductants) [21].

The antioxidant activity and the total antioxidant capacity (TAC) of natural products can be determined using different tests [22], among which the Trolox equivalent antioxidant capacity (TEAC) assays are most frequently used [23]. By virtue of the reaction taking place, these methods are generally distinguished into hydrogen atom transfer (HAT) assays [24] and electron transfer (ET) assays [25,26]. However, the analytical techniques based on the scavenging of stable free radical chromophores (e.g., 2,2′-azinobis-(3-ethylbenzothiazoline-6-sulfonic acid) ABTS, and 2,2-diphenyl1-picrylhydrazyl, DPPH) are classified as mixed-mode tests, since both mechanisms (i.e., ET and HAT) are in place [27]. In turn, the total phenolic contents (TPC) assay based on ET by Folin–Ciocalteu reagent method can measure the reducing capacity of an antioxidant [28].

In each country, hunting practice and traditions may value different attitudes in beaver hunting (food, industry goods). Although, the total quantity of beaver meat entering consumer markets is not recorded, it is important that different products (meat, pelts, and castoreum) still have value for hunters [29]. An interesting part of the beaver carcass is the tail [30] which, especially historically in Europe, was considered a delicacy and prepared like a fish to be eaten during periods of fasting [31]. In North America, beaver tail, feet, and liver are usual indigenous foods even today [32]. The various properties of meat from beaver have previously been investigated, taking into account the microstructure [33], intrinsic properties [34,35], composition of fatty acids [36,37] and amino acids [38,39], concentration of minerals [40], microbiological status [41], and collagen characteristics [42].

To our knowledge, no investigations have been earlier performed to determine the antioxidant status of different tissues of beaver. Moreover, such research on red meat is relatively rarely carried out. Therefore, the current study was undertaken to compare the antioxidant properties including the content of vitamin A (retinol) and vitamin E (α-tocopherol), the total antioxidant activity (ABTS and DPPH free radical-scavenging activity), and the level of total phenolic contents in different tissues of European beaver. Furthermore, the relationships between the chemical composition and antioxidant properties were estimated using principal component analysis (PCA).

## 2. Results

### 2.1. Proximate Composition and Cholesterol Concentration

The percentages of proximate chemical compounds and cholesterol concentration in beaver tissues are presented in Table 1. Significant (*p* < 0.01) variation was found in the percentages of basic composition, with the differences being affected by the nature of the tissues (muscle tissue vs. tail vs. subcutaneous fat). Despite the significant variability between the two adipose tissues compared to muscle tissue, they contained significantly less moisture, protein, and ash while obviously being fatter. In contrast, greater differentiation was observed for cholesterol, the concentration of which did not appear to be related to overall fat content (tail < muscle tissue < subcutaneous fat).

### 2.2. Fat-Soluble Vitamins and Total Antioxidant Capacity

The parameters of total antioxidant capacity for different beaver tissues are shown in Table 2. The concentrations of two fat-soluble vitamins (retinol and α-tocopherol) were significantly affected by type of tissue (adipose tissue vs. muscle tissue). A significantly (*p* < 0.01) higher level of retinol was found in tail and subcutaneous fat compared to muscle tissue from loin and hind leg. Similar results were obtained for α-tocopherol, whereby its concentrations in subcutaneous fat and muscle tissue from loin did not differ significantly.

Among all examined tissues, the tail showed significantly (*p* < 0.01) the highest hydrogen-donating potency (DPPH), the highest capabilities of single electron-transfer (ABTS) and the highest anti-free-radical action (TPC), and also contained the most α-tocopherol. Skeletal muscles (from loin and hind leg) were found to contain significantly (*p* < 0.01) less retinol compared to samples from the tail and subcutaneous fat. In contrast, the latter adipose tissue and muscle tissue had significantly (*p* < 0.01) lower contents of total phenolics. In general, the total antioxidant capacity of the tissues assessed can be ranked as follows (from the highest to the lowest): tail > subcutaneous fat > skeletal muscles.

### 2.3. Correlations and PCA Analysis

Moisture, protein, and ash contents were negatively and significantly correlated with all antioxidant properties (Table 3). The opposite relationships were found for fat content. In contrast, cholesterol was negatively correlated with all indicators of the total antioxidant capacity, although not significantly with retinol. The positive and strong relationships were found between all groups of antioxidant properties, i.e., for concentration of fat-soluble vitamins, results of DPPH and ABTS assays, and between TPC and other indicators of antioxidant capacity.

The principal component analysis (PCA) was made out of 10 variables and 35 cases. The Kaiser criterion (eigenvalues of ≥1) has identified two principal components (PCs) that explained 95.16% of the total variance, as the sum of the first (PC1 = 74.02%) and second (PC2 = 21.13%) principal components (Table 4).

The visualisation of original variables in the two principal components (PC1 × PC2) arrangement is presented in Figure 1. PC1 was positively related to ash, moisture, and protein content, and negatively influenced by ABTS, DPPH, retinol, fat, α-tocopherol, and total phenolics. PC2 was positively related to cholesterol concentration. The coefficients for these correlations are given in detail in Table 4.

The positioning of chemical compounds and indices of the total antioxidant capacity plotted for cases by principal component analysis represented their importance in individual tissues, from which three groupings can be distinguished (Figure 2). Three variables of proximate composition (moisture, ash, and protein content), that were positively correlated with PC1, are placed on the right side of Figure 1 and are representative for skeletal muscles (from loin and hind leg) shown in Figure 2. In turn, ABTS, DPPH, retinol, fat, α-tocopherol, and TPC (the variables negatively correlated with PC1) represent the clearest illustration of the antioxidant potential of tissue from the tail, which is located in the lower left quadrant (Q3, Figure 2). Furthermore, retinol and fat contents were negatively correlated with PC1, corresponding to a combined area of quadrant 3 and 4, in which cases of tissues with the highest content of these compounds (Table 2), i.e., tail (Q3) and subcutaneous fat (Q4) were located. Principal component 2 was most positively correlated with cholesterol content (Figure 1), and the area of this relationship included quadrants 1 and 4, in which cases of subcutaneous fat (Q4) were distributed (Figure 2). In general, the spatial location of individual beaver tissues in the PC1 × PC2 coordinate system proves their mutual variation considering proximate composition and antioxidant potential.

## 3. Discussion

### 3.1. Proximate Composition and Cholesterol Concentration

The results of proximate composition of muscle tissue (from loin and hind leg) are consistent with findings reported previously by other authors for the European beaver from countries bordering the Baltic Sea [33,34,35,38,39,43]. These results indicate very similar levels of moisture, protein, and ash in the meat of wild beavers, but a very variable fat content, which depends in particular on beaver sex [31], biology (diet, habits, physical activity) [44], and season of hunting [37]. Unfortunately, the results for proximate composition of tail and subcutaneous adipose tissue in the beaver are very scarce in the literature. In comparison to the present study, Zalewski et al. [31] have reported a similar content of lipids in skeletal muscles from adult beaver males (0.74% tenderloin and 1.83% thigh), while they found a higher percentage in the tail (66.71%) and a lower percentage in subcutaneous fat (64%). Variability in subcutaneous adipose tissue content has also been shown to be influenced by the time of sampling for analysis. The samples of bovine subcutaneous fat from biopsy contained 67.5% of lipids and 28.8% of moisture, whereas samples taken from the carcass contained 84.9% and 12.9% of fat and moisture, respectively [45].

The *Biceps femoris* of beaver from Latvia contained a very similar level of total cholesterol (49.51 mg/100 g) [39] compared to the present findings for muscle tissue (Table 1), which were also found to be lower compared to previous studies by the authors [36,40] for the hind leg (55–56 mg/100 g). In farmed large semi-aquatic rodents, the lipids percentage and cholesterol content of the muscles were dependent on the confinement system, with higher concentrations found in capybaras without pond management (2.26–4.74% and 52.1 mg/100 g) compared to animals with access to water (1.81–3.93% and 45.7 mg/100 g) [46]. In turn, Pinto et al. [47] reported for meat from free ranging adult capybaras a lipid content between 1.5 and 1.75%, and cholesterol concentration between 35.0 and 47.5 mg/100 g.

### 3.2. Fat-Soluble Vitamins and Total Antioxidant Capacity

The retinol concentrations in the European beaver tissues in the present study were higher than those found for free-ranging reindeer (2.6–3.0 µg/100 g) [48], but considerably lower compared to the range reported for wild boar (69.29–135.16 µg/100 g) [49]. Notably, the concentration of retinol in wild boar *Longissimus dorsi* was conditional on the feeding area (forest vs. farmland) and animal age (1–2 year vs. elder animals).

The α-tocopherol concentrations found in the present study for the European beaver tissues (Table 2) were within the wide range reported by Tejerina et al. [50] for skeletal muscles of Iberian pigs (280–420 µg/100 g) from different production systems and by Realini et al. [51] for beef obtained from cattle fed concentrate (292 µg/100 g) and grazed on pasture (391 µg/100 g). However, our results were higher than values reported by other authors [52,53] for veal (163.88–268 µg/100 g). The higher content of α-tocopherol was found in red deer meat from New Zealand, which ranged from 438 µg/100 g (stags) to 561 µg/100 g (hinds) [54], as well as in meat of reindeer from Norway, amounting to between 498 µg/100 g and 602 µg/100 g [18], and in Iberian deer meat at around 585 µg/100 g [19]. In addition, according to Sampels et al. [48] the high concentration of α-tocopherol in the reindeer meat was related to its non-susceptibility to nonenzymatic oxidation. Nevertheless, the highest concentration of α-tocopherol was reported for the meat of sika deer, 1510 µg/100 g [55], and wild boar, 1740 µg/100 g [56].

The content of α-tocopherol in beef between 300 µg/100 g [57] and 350 µg/100 g [58] is sufficient to limit lipid and heme pigment oxidation. The skeletal muscles of mature male beavers contain rather low levels of intramuscular fat, that amounts to around 1.4% (±0.17%), but with a very high percentage of PUFAs (averaging 49.39%) [36]. In contrast, the total heme pigment content in beaver meat (359–608 ppm) [34] is relatively high compared to red meat from domestic animals [59]. According to Okabe et al. [55] game meats contain high amounts of prooxidants (e.g., phospholipids and myoglobin), substances which enhance the oxidation of lipids. Therefore, the concentration of α-tocopherol in game meat critical to maintaining lipid stability should be at least 700, even up to 900 µg/100 g, i.e., twice as much as, for example, beef and beaver meat. The important role of α-tocopherol as an antioxidant is due to the fact that it is located within cell membranes together with phospholipids [60]. The proportion of this lipid class in the intramuscular fat (from the tenderloin) of beavers is 36.7%, in the subcutaneous fat 15.2% and in the tail 3.1% [31]. In the present study the highest concentration of α-tocopherol was found in the tail, then in the subcutaneous fat, and the lowest in skeletal muscles from the hind leg (Table 2). A similar tendency was observed for retinol; however, its content was dozens of times lower than α-tocopherol.

The contents of α-tocopherol (as vitamin E) and retinol (as vitamin A) in beaver tissues are related to their dietary source. Alpha tocopherol is only synthesised by plants; hence, it is present in the green parts of plants (in chloroplasts) and in plant oils (in concentrated amounts). In animals, α-tocopherol is found in large amounts only in adipose tissue. In animal products, vitamin A occurs essentially as esters of long-chain fatty acids of retinol and pigments (carotenoids). Carotenoids, on the other hand, are synthesised by plants and occur in orange, yellow and green tissues of plants, and a part of them display provitamin A activity (e.g., β-carotene) [61]. The presence of retinol (vitamin A) and other carotenoids (pro-vitamin A) in meat increases the oxidative stability. Carotenoids together with α-tocopherol can scavenge free peroxy radicals, thus effectively limiting oxidation of PUFAs [62]. In different domestic animal species, the variable concentration of α-tocopherol depends on its saturation level in the tissues [63,64]. Animals fed high amounts of fresh green fodder rich in α-tocopherol can accumulate sufficient levels of it in their muscles [65]. In turn, the diet of beavers is highly variable (green and woody parts) and dependent on availability and season (tree parts, herbs, underwater and water plants, agricultural crops) [44]. Moreover, both lipid soluble vitamins (α-tocopherol and β-carotene) display structural similarity, as well as being opponents (antagonistic) because they compete with each other for micellar sites [64].

According to Serpen et al. [16], in order to reliably assess the different mechanisms of the antioxidant potential of a complex matrix such as muscle tissue, it is necessary to perform at least two tests simultaneously. In addition, Mielnik et al. [17] recommend for water soluble antioxidants the oxygen radical absorbance capacity test, and for lipid-soluble antioxidants the assay with DPPH free radical in a polar medium. In the present study to assess the antioxidant capacity of different beaver tissues, two radical assays were used (ABTS and DPPH); total polyphenols were also determined. To date, there is still not enough research concerning the total antioxidant capacity (TAC) of meat of different animal species [66]. Moreover, there are many factors (e.g., analytical) that affect the results of TAC measurement of meat extracts [16].

The contents of total phenolics found in the present study for beaver tissues (Table 2) were substantially higher than the concentrations of TPC reported by Tejerina et al. [50] for the muscles of Iberian pigs, and by Sohaib et al. [67] for breast of broiler chicken. At the same time, our results were considerably lower compared to values reported by Mielnik et al. [17] for reindeer meat. It is well known that plants are the main source of phenolic compounds in human food (fruits, vegetables, herbs, spices, legume seeds, and nuts), which are their most abundant secondary metabolites. Polyphenols can also be found, although in much smaller amounts, in animal tissue, as a consequence of plant feed consumption by animals [68]. However, the intake level of polyphenols from diet does not directly affect their accumulation in animal tissues, details in review [69]. In general, the majority of phenolics are hydrophilic as they contain hydroxyl groups [21]. Some authors pointed out that, unlike tocopherols, phenolic compounds primarily amass in body tissues containing more water, like muscle tissue, than in adipose tissue (back-fat) [68]. Moreover, phenolic compounds (predominantly polyphenols) are probably integrated with mammalian cells by passive transport [70], and their bioavailability is limited due to low lipophilicity, absorption, and fast eradication in the organism [21]. It is however necessary to clarify that polyphenol–lipid interactions in biological systems are more complex [71]. In the present study the highest content of phenolics was found in the tail and, subsequently, in subcutaneous fat and muscle tissue. In this context, it should be noted that the Folin–Ciocalteu assay tends to overestimate the content of phenolic compounds, as it takes into account the involvement in the redox reaction of many other non-phenolic compounds present in the basic medium [66].

Sacchetti et al. [15] reported the total antioxidant activity of chicken meat, on the base of ABTS radical scavenging assay for the hydrophilic and lipophilic extracts, at the level of 0.272 mM TE/100 g for breast, and 0.218 mM TE/100 g for thigh. The cited authors further conclude that the level of the radical scavenging activity of breast meat could be beneficial from a dietary point of view. In contrast, Korzeniowska et al. [72] using a DPPH assay found a higher antioxidative status for muscles from chicken leg (1.467 mM TE/100 g) in comparison with breast (0.609 mM TE/100 g). In turn, Carrillo et al. [73] assessed the total antioxidant capacity of seven fresh meats (breast of chicken or turkey, tenderloin of rabbit, pork or lamb, and fillet of beef or colt) using ABTS assay with the traditional extraction method. While significant differences were found between meats of different species, mean values of TAC ranged from 1.48 mM TE/100 g in beef to 1.87 mM TE/100 g in lamb. In the opinion of Kopec et al. [14], of the three tests used (ABTS, DPPH, and FRAP), the DPPH assay proved to be the most accurate for determining the total antioxidant capacity of turkey meat. Serpen et al. [16] have used the QUENCHER method (as the extraction-independent method) and reported that the total antioxidant capacity of raw meats from domestic animals (beef, chicken, and pork), and fish determined by the ABTS and DPPH assays amounted on average to 1.0 mM TE/100 g, that is two-and-a-half times less than was found in the present study for beaver muscle tissue. Moreover, as also suggested by Serpen et al. [16], numerically lower values of DPPH compared to ABTS results were due to the fact that it is a hydrophobic radical (as opposed to to a hydrophilic ABTS), on the one hand, and it has a greater selectivity for reaction with hydrogen donors, on the other [74]. In turn, Mielnik et al. [17] reported for meat of reindeer from different Norwegian regions an oxygen radical absorbance capacity (ORAC) between 1.62 mM and 1.99 mM TE/100. Taking into account that results were expressed as Trolox equivalent (used as the standard reference), these values were lower than results obtained in the present study. However, it is important to explain that the ORAC test is used to measure the ability of antioxidants to inhibit free peroxyl radicals, which are produced during lipid oxidation [75]. It is also worth mentioning that Sacchetti et al. [15] showed significantly higher total antioxidant activity in chicken breast muscles compared to thigh muscles, which, according to the authors, was influenced by the higher lipid content and fat-soluble antioxidant compounds in the hind legs. In contrast, hydrophilic compounds showing higher activity predominated in lean breast meat. Nevertheless, such a relationship was not observed in the present study on beaver muscles with different fat contents (loin vs. hind leg).

### 3.3. Correlations and PCA Analysis

In the present research, significant positive correlations were obtained between individual properties of antioxidant activity which, in turn, were negatively correlated with proximate compounds with the exception of fat (Table 3). Our results are in line with the previous findings of Tejerina et al. [50] in regard to the close relationship between antioxidant contents (tocopherols and total phenolics) and antioxidant (lipophilic and hydrophilic) activities of muscle tissue from Iberian pigs. Moreover, Mielnik et al. [17] found for reindeer meat significant correlations between TPC and antiradical potential (r = 0.686; *p* < 0.001) and oxygen radical absorbance capacity (r = 0.433; *p* = 0.013). At the same time there was no relationship between these radical assays (ARP and ORAC), whereas in the present study ABTS and DPPH were very closely correlated (Table 3).

The use of multivariate principal component analysis (PCA) made it possible to visualise the natural differences in the antioxidant activity of different tissues of the European beaver, and at the same time allowed them to be grouped together. A high degree of similarity was shown between muscle tissues from different carcass locations. In turn, Carillo et al. [73], using PCA analysis, determined two principal components that explained a total of 79.8% of the overall variability for the ABTS test results. At the same time, the authors pointed out that meat colour was the differentiating factor for antioxidant activity, grouping red meat (lamb, beef, and colt) vs. white meat (chicken, turkey, rabbit, and pork). In addition, in the present study it was revealed that the two tissues with a high lipid content, i.e., from the tail and the adipose subcutaneous layer, were distant from each other, which was associated with significantly different polyphenol content.

Summing up, the obtained results suggest that antioxidant activity tended to be associated with lipophilic (hydrophobic) antioxidants, which were positively correlated with total tissue lipid levels and negatively linked to cholesterol concentration. Furthermore, it is to be presumed that the antioxidant potential of beaver tissues consists of other mechanisms in addition to lipophilic components [76]. It has been confirmed previously that proteins and peptides show potent antioxidant activity in muscle tissue by scavenging free radicals and binding prooxidant metal ions [77]. In contrast, the high antioxidant activity of subcutaneous fat despite its low total polyphenol concentration may have been compensated for by the greater presence of vitamins. Furthermore, it is worth mentioning that recently Zhang et al. [78] indicate that the North American beaver may owe its longevity and cancer resistance to evolutionary changes in gene expression. These changes include a greater ability to detoxify large amounts of aldehydes in the beaver body. This state is the result of aldehyde abundance in the specific beaver’s woody feed (from bark and cambium) and as the outcome of lipid oxidation. Indeed, a high proportion of polyunsaturated fatty acids (PUFAs) was found in the lipids of various beaver tissues, which is remarkable in comparison to other mammals [31,36]. In a previous study Florek et al. [34] reported for the muscle tissue of the beaver a relatively low initial (24 h postmortem) Tbars value (0.15 mg MDA/kg), which significantly increased threefold after 7 days (0.46 mg MDA/kg). Nevertheless, this is still a low level for this indicator of secondary oxidation of PUFAs [12].

## 4. Materials and Methods

### 4.1. Animals and Sampling

The research material was made up of carcasses of European beavers (*Castor fiber* L.) provided by approved hunters on the basis of two grants of permission (WPN.6401.33.2016.KM of 7 Apr 2016 and WPN.6401.85.2016.KM of 4 July 2016) and the decision of the Regional Director for Environmental Protection (the Official Journal of Lublin voivodeship of 1 December 2016, item 4828). All the details of the animals, and the collection and storage of the samples have been described previously [41,42]. In the present study carcasses of nine adult males were investigated. The *Longissimus thoracis* and *Longissimus lumborum* muscles were excised from the loin (n = 9), and *Semimembranosus*, *Biceps femoris* and *Semitendinosus* muscles were excised from the hind leg (n = 9), and then samples were prepared in accordance with Domaradzki et al. [36]. The samples of other tissues included subcutaneous fat (n = 9) and the inner part of tail (n = 8). The beaver’s tail is flat, paddle-shaped, and covered with keratinised scales of a hexagonal pattern.

### 4.2. Chemical Analyses

#### 4.2.1. Chemical Compounds Studied in This Article

*n*-Hexane (PubChem CID: 8058); methanol (PubChem CID: 887); potassium hydroxide (PubChem CID: 14797); ethanol (PubChem CID: 702); 6-ketocholestanol (PubChem CID: 102008); acetonitrile (PubChem CID: 6342); isopropanol (PubChem CID: 3776); sodium carbonate (PubChem CID: 10340); gallic acid (PubChem CID: 370); Trolox (PubChem CID: 40634); DPPH (PubChem CID: 2735032); ABTS (PubChem CID: 9570474); potassium persulfate (PubChem CID: 24412); ethyl acetate (PubChem CID: 8857); methyl tert-butyl ether (PubChem CID: 15413); all-trans-retinol (PubChem CID: 445354); D-alfa-tocopherol (PubChem CID: 14985).

#### 4.2.2. Proximate Composition and Cholesterol Content

The proximate composition of samples of muscle and adipose tissues was determined as earlier described in detail by Florek et al. [42]. Determination of cholesterol concentration was conducted in two steps. Samples were saponificated (first step) according to Stewart et al. [79] with modification. Briefly, the sample (2 g) was mixed with 4 mL potassium hydroxide (50%) and 6 mL ethanol (95%) until complete dissolution at 40 °C, and then heated for 10 min at 60 °C. Following cooling with 5 mL water, the unsaponifiable fraction was extracted with *n*-hexane three times using 10 mL, then 3 mL aliquots of extract were dried under a purified nitrogen stream. The resulting saponification samples were analysed by using the HPLC method (second step). Extracts were dissolved in 3 mL acetonitrile and isopropanol solution (70:30, *v*/*v*), and finally 1 mL was used for HPLC analysis [80]. Shimadzu HPLC equipment was used, consisting of three-component solvent delivery system (LAD 10), the Rheodyne loop injector (20 µL, column temperature 30 °C), a UV-visible detector, and CLAS-VP 10 software. The analysis was carried out using the Lichrospher 5RP18 column (150 mm × 4.6 mm), and acetonitrile and isopropanol (70:30, *v*/*v*) were used as mobile phase at a flow rate of 1 mL/min. Detection was carried out at a wavelength of 210 nm. The cholesterol was identified by comparison of sample retention time with standard (C8667, Sigma-Aldrich, St. Louis, MO, USA). The internal standardisation (0.504 mg 6-ketocholestanol, K1250, Sigma-Aldrich, St. Louis, MO, USA) after saponification was used for quantification of each sample.

#### 4.2.3. Properties of Antioxidant Activity

##### Preparation of Extracts

Extracts were prepared by weighing 0.5 g of lyophilisate (Freeze Dryer Alpha 1–4 LSC, Christ, Osterode, Germany) and dissolving in 4 mL of methanol (Avantor-POCh, Poland) and water mixture (50:50, *v*/*v*), then the solution was shaken for 1 h and centrifuged at 1800× *g* for 3 min. Ready-made extracts were tightly sealed and stored at +4 °C until total phenolic content and free radical scavenging activity determinations. The weight of samples was determined before and after freeze drying in order to calculate dry weight into wet weight basis.

##### Total Phenolic Content

The total phenolic content (TPC) was determined as a result of the reaction of 0.1 mL extracted sample and 0.9 mL distilled water (HLP 20UV, HYDROLAB, Straszyn, Poland) with 0.1 mL Folin–Ciocalteu reagent [81]. After 5 min, 1 mL of 7% Na_2_CO_3_ and 0.4 mL distilled water were added separately and mixed. Following 50 min of incubation in the dark (20 °C), the absorbance was measured at λ = 765 nm using a UV-2600i spectrophotometer (Shimadzu, Tokyo, Japan) with respect to a blank test. A calibration curve was plotted for a standard solution of gallic acid (GAE, Sigma-Aldrich, Munich, Germany) between 0 and 10 mg/100 mL, and total phenolic content was expressed as gallic acid equivalent (GAE) in mg per 100 g of wet sample (mg GAE/100 g).

#### 4.2.4. Free Radical Scavenging Activity

##### DPPH Method

Free radical scavenging activity was determined using 0.08 mL of extracted sample and 3.92 mL of DPPH• (2,2-diphenyl-1-picrylhydrazyl, Sigma Aldrich Co., St. Louis, MO, USA) methanolic solution (concentration 6 × 10^−5^ mol/L). The reaction mixture was incubated in the dark (20 °C) for 30 min and absorbance was recorded at λ = 515 nm (UV-2600i spectrophotometer, Shimadzu, Tokyo, Japan). The calibration curve was plotted using Trolox (Sigma-Aldrich, Munich, Germany) ethanolic solutions (1 mM) at a concentration ranging from 0 to 2.5 mg/100 mL, and results were expressed as millimole equivalent of Trolox per 100 g of wet sample (mM TE/100 g).

##### ABTS Method

The ability of the extracts to eliminate free radicals was determined by Re et al. [26] using 2,2-azinobis-(3-ethylbenzothiazoline-6-sulphonic) acid (Sigma-Aldrich, Munich, Germany). The stock solution of ABTS• (7 mM/L ABTS, 2.45 mM/L K_2_S_2_O_8_) was diluted using PBS (phosphate buffered saline, pH 7.4) to ensure an absorbance at λ = 734 nm was between 0.74 and 0.75 (UV-2600i spectrophotometer, Shimadzu, Tokyo, Japan), and prepared working solution was stored in a glass-stoppered dark bottle. The reaction mixture was incubated for 6 min at 30 °C and the absorbance was measured at λ = 734 nm. The calibration curve was plotted using Trolox (Sigma-Aldrich, Munich, Germany) solutions (1 mM) in PBS at a concentration range between 0 and 10 mg/100 mL, and results were expressed as millimole equivalent of Trolox per 100 g of wet sample (mM TE/100 g).

#### 4.2.5. Alpha Tocopherol and Retinol Contents

Freeze-dried sample (2 g) was mixed with 1 mL distilled water, then 4 mL ethanol:hexane solution (1:1) was added. The organic phase was separated by means of Pasteur pipette following centrifugation at 1800× *g* for 10 min at 4 °C. The aqueous phase was extracted twice with *n*-hexane. Both extracts were evaporated until dry using nitrogen stream. The dry residue was dissolved in 50 μL of ethyl acetate until HPLC analysis. The content of selected fat-soluble vitamins was measured by means of ProStar 240 HPLC System (Varian, Inc., Walnut Creek, CA, USA) equipped with photodiode array detector, quaternary solvent delivery system, temperature control module set at 20 °C, automatic injector (20 μL), and Hypersil ODS C18 column (150 mm × 4.6 mm, 5 μm). The mobile phase comprised methanol (MeOH), methyl tert-butyl ether (MTBE), and water, added according to the following elution parameters by Mouly et al. [82]: 0 min: 90% MeOH + 5% MTBE + 5% water; 12 min: 95% MeOH + 5% MTBE; 25 min: 89% MeOH + 11% MTBE; 40 min: 75% MeOH + 25% MTBE; 50 min: 40% MeOH + 60% MTBE; 56 min: 15% MeOH + 85% MTBE; 62 min: 90% MeOH + 5% MTBE + 5% water. The flow rate of the mobile phase was 1 mL/min. Detection was carried out at 325 nm for retinol and at 280 nm for α-tocopherol. Vitamin A and vitamin E were identified using standards, i.e., all-trans-retinol and D-α-tocopherol, respectively (Sigma-Aldrich, Madrid, Spain), which were at least 90% pure, and quantified on the basis of the calibration curve of standard solutions. The weight of samples was determined before and after freeze drying in order to calculate dry weight into wet weight basis. Results were expressed as µg per 100 g of fresh tissue.

### 4.3. Statistical Analysis

All measurements were carried out in duplicate and the results were analysed statistically using STATISTICA 13 (TIBCO Software Inc., Palo Alto, CA, USA). One-way analysis of variance (ANOVA) was used to determine the fixed effect of beaver tissue on the proximate composition and parameters of antioxidant activity. Tukey’s HSD test was applied to determine the significance of differences between means, and the statistical significance was set at *p* < 0.05 or *p* < 0.01. In tables mean value and standard error are given. The coefficient of Pearson correlation for particular pairs of variables was calculated. In order to characterise the inter-tissue differentiation, the principal component analysis (PCA) was further applied to visualise data and demonstrate the relationships between tested variables.

## 5. Conclusions

Taking into account the obtained results, it can be concluded that the type of tissue significantly affected content of all chemical components. The concentration of cholesterol was not related to total fat content (tail < muscle tissue < subcutaneous fat). The retinol and α-tocopherol contents were significantly higher in adipose tissue from the tail and subcutaneous fat compared to skeletal muscles. Among all tissues the tail showed significantly the highest values of DPPH and ABTS tests, and contained the most polyphenols. It was proven that moisture, ash, and protein contents were representative for skeletal muscles, whereas, α-tocopherol, phenolics, ABTS, and DPPH accurately described the antioxidant capacity of tissue from the tail. Given the small number of investigations in this scope and the limited number of results, its continuation seems justified. The obtained results may therefore constitute a valuable supplement to public databases on content and food nutritive value, and serve as comparative data.

## Figures and Tables

**Figure 1 molecules-27-08973-f001:**
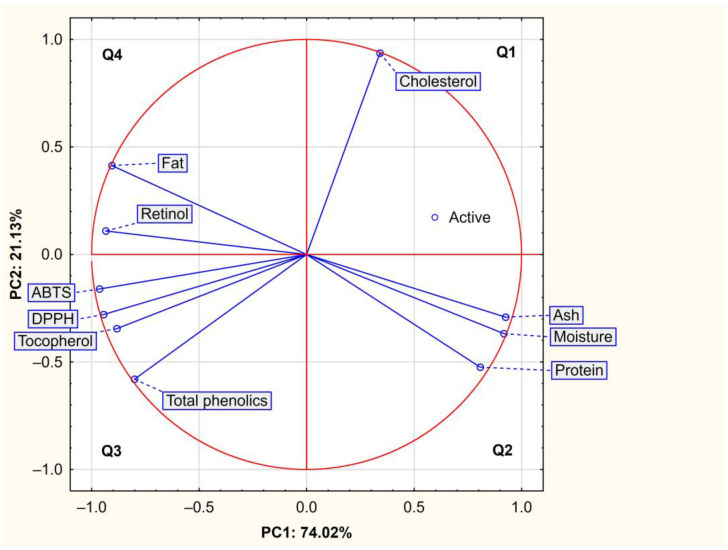
Projection of variables in a two-factor plane (PC1 × PC2).

**Figure 2 molecules-27-08973-f002:**
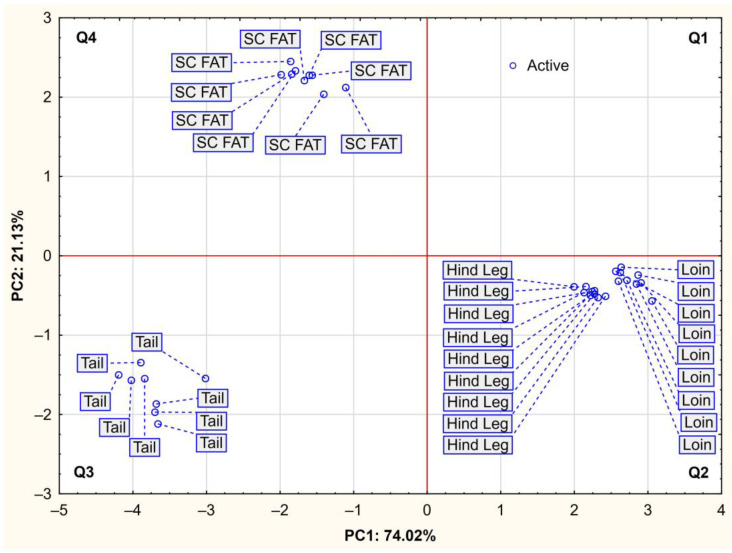
Projection of cases depending on type of beaver tissue in the a two-factor plane (PC1 × PC2); SC Fat, subcutaneous fat; Loin: *Longissimus lumborum* and *Longissimus thoracis*; Hind Leg: *Semimembranosus*, *Biceps femoris* and *Semitendinosus*.

**Table 1 molecules-27-08973-t001:** Proximate composition (%) and cholesterol content (mg/100 g) of beaver tissues (mean and standard error).

Compound	Loin (*Longissimus thoracis* and *Longissimus lumborum*)	Hind Leg (*Semimembranosus, Biceps femoris* and *Semitendinosus*)	Tail	Subcutaneous Fat
Moisture	76.16 ^C^ ± 0.14	75.75 ^C^ ± 0.20	25.37 ^B^ ± 3.28	16.46 ^A^ ± 0.91
Protein	22.13 ^C^ ± 0.17	21.86 ^C^ ± 0.29	11.65 ^B^ ± 1.58	6.22 ^A^ ± 0.55
Fat	0.71 ^A^ ± 0.10	1.54 ^A^ ± 0.32	61.69 ^B^ ± 2.97	76.62 ^C^ ± 1.11
Ash	1.13 ^B^ ± 0.07	1.12 ^B^ ± 0.05	0.26 ^A^ ± 0.03	0.23 ^A^ ± 0.02
Cholesterol	49.76 ^BC^ ± 0.07	49.17 ^AB^ ± 0.04	44.87 ^A^ ± 0.03	52.85 ^C^ ± 0.05

Means in the same row with different superscripts ^A, B, C^ are significantly different at *p* < 0.01.

**Table 2 molecules-27-08973-t002:** Concentration of retinol, α-tocopherol, and antioxidant activity of beaver tissues (mean and standard error).

Compound	Loin (*Longissimus thoracis* and *Longissimus lumborum*)	Hind Leg (*Semimembranosus, Biceps femoris* and *Semitendinosus*)	Tail	Subcutaneous Fat
Retinol (µg/100 g)	8.61 ^A^ ± 0.14	9.56 ^A^ ± 0.13	13.02 ^B^ ± 0.10	12.23 ^B^ ± 0.43
α-Tocopherol (µg/100 g)	366.29 ^B^ ± 0.39	356.59 ^A^ ± 1.35	391.16 ± 0.23	371.32 ^BC^ ± 0.21
Total phenolics (mg GAE/100 g)	150.94 ^A^ ± 1.10	167.33 ^A^ ± 1.55	543.69 ^B^ ± 23.98	186.29 ^A^ ± 4.35
DPPH (mM TE/100 g)	2.34 ^A^ ± 0.02	2.51 ^AB^ ± 0.06	3.07 ^C^ ± 0.07	2.68 ^BC^ ± 0.01
ABTS (mM TE/100 g)	2.58 ^A^ ± 0.02	2.78 ^AB^ ± 0.04	3.33 ^C^ ± 0.03	3.01 ^BC^ ± 0.05

Means in the same row with different superscripts ^A, B, C^ are significantly different at *p* < 0.01.

**Table 3 molecules-27-08973-t003:** Coefficients of Pearson’s correlation (n = 35).

Variable	Retinol	α-Tocopherol	Total Phenolics	DPPH	ABTS
Moisture	−0.885 **	−0.692 **	−0.539 **	−0.754 **	−0.815 **
Ash	−0.880 **	−0.718 **	−0.578 *	−0.787 **	−0.831 **
Fat	0.873 **	0.669 **	0.497 *	0.733 **	0.800 **
Protein	−0.777 **	−0.552 **	−0.330	−0.617 **	−0.697 **
Cholesterol	−0.215	−0.621 **	−0.810 **	−0.586 **	−0.481 *
Retinol	–	0.748 **	0.666 **	0.858 **	0.894 **
α-Tocopherol	–	–	0.913 **	0.886 **	0.851 **
Total phenolics	–	–	–	0.907 **	0.852 **
DPPH	–	–	–	–	0.989 **

* *p* < 0.01, ** *p* < 0.001.

**Table 4 molecules-27-08973-t004:** Eigenvalues, the proportion of variation (%), and correlations between the principal components and the original variables.

Component	PC1	PC2
Eigenvalue	7.40	2.11
Proportion of variance (%)	74.02	21.13
Cumulative of variance (%)	74.02	95.16
Variable		
Moisture (%)	0.917	−0.368
Protein (%)	0.807	−0.525
Fat (%)	−0.906	0.413
Ash (%)	0.925	−0.291
Cholesterol	0.341	0.936
Retinol	−0.935	0.109
α-Tocopherol	−0.883	−0.345
Total phenolics	−0.801	−0.580
DPPH	−0.944	−0.280
ABTS	−0.964	−0.160

## Data Availability

Not applicable.

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
