# Peer review of "Antioxidant Properties and Proximate Composition of Different Tissues of European Beaver"

_molecules, 2022, doi:10.3390/molecules27248973_

Round 1

Reviewer 1 Report

The authors developed an interesting study about determination of the antioxidant status of different tissues of European beaver. The study involved estimation of the content of vitamin A (retinol) and vitamin E (α-tocopherol), the total antioxidant activity (ABTS and DPPH free radical-scavenging activity), and the level of total phenolic contents in different tissues of European beaver.  In addition, they used the principal component analysis (PCA) to measure the relationships between the chemical composition and antioxidant properties. The manuscript is written well, and the discussion of the analytical methods (spectrophotometry and HPLC) employed for the analysis of the total phenolic contents and free radical scavenging activity (DPPH and ABTS assays), as well as fat-soluble vitamins was performed efficiently.

Herein, I recommend accepting the manuscript in its current form.  

Author Response

Reviewer 1

Comments and Suggestions for Authors

 The authors developed an interesting study about determination of the antioxidant status of different tissues of European beaver. The study involved estimation of the content of vitamin A (retinol) and vitamin E (α-tocopherol), the total antioxidant activity (ABTS and DPPH free radical-scavenging activity), and the level of total phenolic contents in different tissues of European beaver.  In addition, they used the principal component analysis (PCA) to measure the relationships between the chemical composition and antioxidant properties. The manuscript is written well, and the discussion of the analytical methods (spectrophotometry and HPLC) employed for the analysis of the total phenolic contents and free radical scavenging activity (DPPH and ABTS assays), as well as fat-soluble vitamins was performed efficiently.

Herein, I recommend accepting the manuscript in its current form. 

Answer. Thank you for very kind statement and sympathetic recommendation.

Reviewer 2 Report

It is a research that characterizes the antioxidant properties of different beaver tissues; however, in my opinion, it is too short and it does not provide more information than the characterization of two compounds that are widely known to have antioxidant activity. But for the quality of the journal, they must carry out at least one shelf-life test (oxidative stability of color, lipids, proteins) of the different beaver tissues, which demonstrate how these compounds with antioxidant activity interact with complex matrices such as meat and lipid.

L20: Misspelling, change 'lion' to 'loin'.

L56: There are too many citations (14-21), I recommend leaving only those from the last 10 years.

L111-112: In Table 2 you must homogenize the same number of decimal places (2) in all values.

 L358: I suggest that you remove quote 41 and put the two grants of permission.

 L358-360: I suggest that you specify the hunting season and the year in which the animals used in this study were hunted. Also, explain the handling they gave to the meat after slaughter (how long did the meat last from hunting to the antioxidant activity analysis), how were the conservation conditions (refrigerated, frozen or packaged) and freeze-drying of the meat.

 L360-362: after the separation of the loin or hind leg muscles, were they mixed? explain if they were homogenized to form a single mixture of each group or each muscle was analyzed separately?

 L410-411; L420-421; L431-432: It is necessary that you put the concentration range that you used for the calibration curve.

 L455: in this section you must also specify that Pearson correlations were performed.

 L456: Explain why they only duplicated and not tripled each measurement?

 L461: Explain why they propose two values of statistical significance (p < 0.05 or p < 0.01).

 L472-474: "allowed identifying two principal components which explained 95.16% of the total variance, and grouped together tissues according to their similarity". This is not a conclusion, it is a repetition of the results. I suggest removing it from this section and correcting the wording.

 L476-478: "Perhaps this is related to the specific physiological role played by the tail in beavers, as it is involved in fat deposition and thermoregulation of the body." This is discussion, not conclusion. Change it to the discussion section.

L480-481: Based on the conclusions you present about the antioxidant properties found in the tail, I ask if the tail is also consumed. If it is affirmative, they should have a small section on the edible parts of the beaver in the introduction.

Author Response

Dear Reviewer,

The authors would like to warmly thank you for all comments and suggestions, especially the critical ones, aimed at improving the scientific value of the article and eliminating the most important errors. We greatly appreciate the opportunity that we have been given to further revise the manuscript. We believe that you will share the arguments submitted by authors and find this revision fully satisfactory.

Reviewer 2

Comments and Suggestions for Authors

It is a research that characterizes the antioxidant properties of different beaver tissues; however, in my opinion, it is too short and it does not provide more information than the characterization of two compounds that are widely known to have antioxidant activity. But for the quality of the journal, they must carry out at least one shelf-life test (oxidative stability of color, lipids, proteins) of the different beaver tissues, which demonstrate how these compounds with antioxidant activity interact with complex matrices such as meat and lipid.

Answer. Thank you for this valuable opinion. We wanted to notify that some of the suggested properties of muscle tissue (Tbars, colour) have been assessed during refrigerated storage (Florek et al. 2017), as well as citing some results in this paper (L351-353). However, due to the nature of the study material (possibility of collection and quantity), conducting a broad study is very limited. Of course, we thank you for directing further research should the opportunity to do so arise.    

 L20: Misspelling, change 'lion' to 'loin'.

Answer. Thank you for this remark, this mistake was corrected.

L56: There are too many citations (14-21), I recommend leaving only those from the last 10 years.

Answer. Thank you for this remark, the number of references has been limited to the most important literature.

L111-112: In Table 2 you must homogenize the same number of decimal places (2) in all values.

Answer. Thank you for this suggestion, values have been unified.

L358: I suggest that you remove quote 41 and put the two grants of permission.

Answer. Thank you for this suggestion, the requested information has been completed.

L358-360: I suggest that you specify the hunting season and the year in which the animals used in this study were hunted. Also, explain the handling they gave to the meat after slaughter (how long did the meat last from hunting to the antioxidant activity analysis), how were the conservation conditions (refrigerated, frozen or packaged) and freeze-drying of the meat.

Answer. Thank you for this suggestion, however, we would like to inform that all details of the animals, sample collection and storage have been described previously in papers Ziomek et al. (2021) and Florek et al. (2022). On the one hand, the authors wished to avoid repeating the same information and, on the other, they wanted to avoid unnecessarily increasing the similarity with the earlier work, which is now being highlighted by the editors.  Therefore, in the current version of the manuscript, we have restricted ourselves to indicating two previously mentioned papers.

L360-362: after the separation of the loin or hind leg muscles, were they mixed? explain if they were homogenized to form a single mixture of each group or each muscle was analyzed separately?

Answer. Thank you for this question. Indeed, pooled samples were created from individual skeletal muscles from the dorsal part and hind limb of each carcass. 

L410-411; L420-421; L431-432: It is necessary that you put the concentration range that you used for the calibration curve.

Answer. Thank you for this request, the missing information was completed for all the analyses specified.

L455: in this section you must also specify that Pearson correlations were performed.

Answer. Thank you for this remark, the missing information about Pearson correlations was completed.

L456: Explain why they only duplicated and not tripled each measurement?

Answer. Thank you for this remark. Certainly, we are aware that for chemical analysis is better to perform more replications for a sample (three than two). However, all the analyses presented in the manuscript in our laboratory are routinely performed for many years, we are sure that they are reliable, robust and are under control. Therefore, in a routine analysis, two replicates per measurement is an optimal and acceptable effort for us in terms of exploiting the capacity of team (as University teachers), time and foremost financial resources.

L461: Explain why they propose two values of statistical significance (p < 0.05 or p < 0.01).

Answer. Thank you for this remark. We would like to inform that in our studies we routinely adopt 2 levels of significance. Since in our research (within agricultural science), an estimation error of 5 % is usually assumed, and if there are significant differences, we reduce the error to 1 %. We use lowercase (p<0.05) or uppercase (p<0.01) letters to denote different levels of significance. If this way is not clear, of course, the authors will adapt to the recommendations. We mention that we have corrected the marking in Table 3.

L472-474: "allowed identifying two principal components which explained 95.16% of the total variance, and grouped together tissues according to their similarity". This is not a conclusion, it is a repetition of the results. I suggest removing it from this section and correcting the wording.

Answer. Thank you for this suggestion, and this critical remark has been taken into account by the authors.

L476-478: "Perhaps this is related to the specific physiological role played by the tail in beavers, as it is involved in fat deposition and thermoregulation of the body." This is discussion, not conclusion. Change it to the discussion section.

Answer. Thank you for this suggestion. This disputable sentence was deleted.

L480-481: Based on the conclusions you present about the antioxidant properties found in the tail, I ask if the tail is also consumed. If it is affirmative, they should have a small section on the edible parts of the beaver in the introduction.

Answer. Thank you for this remark, however, in view of the current size of the beaver population, the limited harvest of this species and thus the low level of consumption restricted basically to interested hunters, the present study is exploratory. Nevertheless, the introduction was supplemented with information on the edible parts, especially the tail, as the muscular tissue of beavers is quite well characterised.

Reviewer 3 Report

Comments on paper

Antioxidant properties and proximate composition of different tissues of European beaver

The authors of the present study investigated the composition of several tissues from wild beavers, content of some antioxidants and free radical scavenging activities. The study has been well carried out and some data may be interesting also for nutritionists who investigate domestic animals.

·      Beavers were provided by hunters; thus various data are not available. Nevertheless, the authors probably know the weight and sex of beavers and rough time of slaughter. Some information may be obtained post-mortem, e.g. ratio of green and woody parts of diet.

·      Check the order of chapters in your manuscript. Materials and Methods should follow Introduction.

·      L. 56: The nonenzymatic endogenous antioxidants are primarily bilirubin and uric acid, but not vitamins E and C. Vitamins must be obtained by diet.

Conclusion: Moderate revision is necessary.

Author Response

Dear Reviewer,

The authors would like to warmly thank you for all comments and suggestions, especially the critical ones, aimed at improving the scientific value of the article and eliminating the most important errors. We greatly appreciate the opportunity that we have been given to further revise the manuscript. We believe that you will share the arguments submitted by authors and find this revision fully satisfactory.

Reviewer 3

Comments and Suggestions for Authors

Comments on paper

Antioxidant properties and proximate composition of different tissues of European beaver

The authors of the present study investigated the composition of several tissues from wild beavers, content of some antioxidants and free radical scavenging activities. The study has been well carried out and some data may be interesting also for nutritionists who investigate domestic animals.

Answer. Thank you for this kind statement.

Beavers were provided by hunters; thus various data are not available. Nevertheless, the authors probably know the weight and sex of beavers and rough time of slaughter. Some information may be obtained post-mortem, e.g. ratio of green and woody parts of diet.

Answer. Thank you for this remark. We would like to inform that all details of the animals, sample collection and storage have been described previously in papers Ziomek et al. (2021) and Florek et al. (2022). On the one hand, the authors wished to avoid repeating the same information and, on the other, they wanted to avoid unnecessarily increasing the similarity with the earlier work, which is now being highlighted by the editors.

Check the order of chapters in your manuscript. Materials and Methods should follow Introduction.

Answer. Thank you for this remark, however, the authors prepared the article strictly according to the editorial requirements using the original Molecules template.

  1. 56: The nonenzymatic endogenous antioxidants are primarily bilirubin and uric acid, but not vitamins E and C. Vitamins must be obtained by diet.

Answer. Thank you very much for this valuable suggestion, this disputable sentence was corrected.

Conclusion: Moderate revision is necessary.

Answer. Thank you for this suggestion, authors have redrafted this part of the article in line with the recommendations of other reviewers.

Round 2

Reviewer 2 Report

I appreciate that you have taken into account the observations and suggestions made to the manuscript, with this it was considerably improved.